# Time to Onset of Paresthesia Among Community Members Exposed to the World Trade Center Disaster

**DOI:** 10.3390/ijerph16081429

**Published:** 2019-04-22

**Authors:** Sujata Thawani, Bin Wang, Yongzhao Shao, Joan Reibman, Michael Marmor

**Affiliations:** 1Department of Neurology, New York University School of Medicine, New York, NY 10017, USA; sujata.thawani@nyulangone.org; 2Department of Population Health, New York University School of Medicine, New York, NY 10016, USA; bin.wang@nyulangone.org (B.W.); yongzhao.shao@nyulangone.org (Y.S.); 3Department of Environmental Medicine, New York University School of Medicine, New York, NY 10016, USA; joan.reibman@bellevue.nychhc.org; 4Department of Medicine, New York University School of Medicine, New York, NY 10016, USA

**Keywords:** paresthesia, neuropathic symptoms, Cox regression, hazard function, World Trade Center exposure

## Abstract

We examined whether time to onset of paresthesia was associated with indicators of severity of World Trade Center (WTC) exposure. We analyzed data from 3411 patients from the Bellevue Hospital—WTC Environmental Health Center. Paresthesia was defined as present if the symptom occurred in the lower extremities with frequency “often” or “almost continuous.” We plotted hazard functions and used the log-rank test to compare time to onset of paresthesia between different exposure groups. We also used Cox regression analysis to examine risk factors for time-to-paresthesia after 9/11/2001 and calculate hazard ratios adjusted for potential confounders. We found significantly elevated hazard ratios for paresthesia for (a) working in a job that required cleaning of WTC dust in the workplace; and (b) being heavily exposed to WTC dust on September 11, 2001, after adjusting for age, race/ethnicity, depression, anxiety, post-traumatic stress disorder, and body mass index. These observational data are consistent with the hypothesis that exposure to WTC dust or some other aspect of cleaning WTC dust in the workplace, is associated with neuropathy and paresthesia. Further neurological evaluations of this and other WTC-exposed populations is warranted.

## 1. Introduction

The World Trade Center (WTC) Environmental Health Center (EHC) serves community members exposed to the World Trade Center disaster of September 11, 2001. These individuals include local workers and residents, as well as other affected community members with potential exposure to WTC contaminants who developed physical or mental health symptoms after the event. The most common presenting complaints are aerodigestive symptoms [1,2]. However, within a few years after 9/11, patients seeking care at the Bellevue Hospital WTC EHC also reported paresthesias or sensations of tingling, prickling, or numbness in the extremities. Formal questions about paresthesias thus were added to the medical evaluation questionnaires around January 1, 2008 to evaluate for the potential for peripheral neuropathy associated with WTC toxic exposures. 

Neuropathic symptoms were associated with sarin exposure that occurred during the terrorist attacks in Japan in 1994 and 1995 [3]. Of 155 individuals who required medical evaluation, 40 subjects required hospital admission due to severe exposure. These subjects had more frequent salivation, rhinorrhea, and diarrhea. 40% (16/40) of these subjects reported distal lower extremity dysesthesias and were noted to have reduced distal vibration on clinical exam. These signs and symptoms resolved at 3 months follow-up in the majority of cases.

Components of the WTC toxins provide biologic plausibility for the development of neuropathic disease. Heavy metals and environmental toxin exposures have been associated with distal symmetric polyneuropathy [4]. Examination of sciatic nerves from rats exposed to WTC dust also demonstrated reductions in conduction velocities of nerve action potentials when compared to controls [5]. WTC dust contained known neurotoxins including lead and complex hydrocarbons [6,7,8,9]. Workers involved in clean-up activities may also have been exposed to organic solvents. Distal sensory motor neuropathy is associated with lead poisoning, with a motor predominant neuropathy primarily seen following acute exposure [10]. Industrial exposure to n-hexane, an organic solvent, has been associated with paresthesia and distal symmetric, primarily axonal, large fiber polyneuropathy [11,12]. 

An earlier analysis of patients from the WTC EHC found that many reported paresthesias at enrollment 7–15 years after 9/11 [13]. These symptoms were associated with the severity of exposure to WTC dust on 9/11 and cleaning of workplaces after adjusting for known risk factors for paresthesia, including preexisting paresthesias, anxiety, and diagnoses of diabetes and cancer [3,12]. Paresthesias also were associated with respiratory symptoms and reduced lung function, which may have been surrogate measures of WTC exposure [4,14]. Paresthesia symptoms were associated with later (2013–2015) enrollment in the WTC EHC, suggesting a long latency period between exposure and development of symptoms in some patients. Neuropathic symptoms also were reported in a series of 16 first responders exposed to the WTC disaster [15]. 56% (9/16) of these patients had neuropathy that could not be explained by diabetes, B12 deficiency, or Lyme disease suggesting WTC exposure as a possible etiology. The present analyses aim to describe further paresthesia associated with WTC exposure by exploring time to paresthesia after the September 11, 2001 disaster and associations of time to paresthesia with measures of severity of WTC exposure. 

## 2. Materials and Methods 

### 2.1. Study Population

As previously described, we enrolled subjects for the present analysis from the Bellevue Hospital WTC Environmental Health Center (WTC EHC) [2]. Enrollment into the WTC EHC was open to individuals who were exposed to the disaster who reported medical or mental health symptoms certified as potentially due to exposure to the WTC event [2]. Paresthesias were not prerequisite symptoms for inclusion in the program. Patients of the WTC EHC primarily included individuals who worked in local businesses, lived in buildings near the WTC towers, or were traveling through the area of Manhattan south of Canal Street on the day of the disaster. A small proportion of the WTC EHC patient population were first-responders who were enrolled during the initiation of the program. Subsequently, separate programs were developed for first responders and community members [2]. Blood glucose was measured by the Bellevue Hospital Clinical Laboratory. Body mass index (BMI), post-traumatic stress disorder (PTSD), depression and anxiety were measured as previously described [16]. We limited the present analyses to subjects who responded to questions added around January 1, 2008 regarding paresthesia.

The New York University School of Medicine Institutional Review Board approved the research database used for these analyses. The present analyses used only data from patients who provided informed consent to allow the use of their data in research analyses.

### 2.2. Paresthesia

We investigated the incidence of paresthesia of the lower extremities. We defined paresthesia to be a positive response to the question, “In the past year, did you experience a prickling or tingling feeling, with or without an asleep feeling, in your feet or legs?” Moreover, we further limited persons with paresthesia to those who responded that they experienced this feeling “often” or “almost continuous”. Persons who said they experienced this feeling “occasionally” or “never” were classified as not having paresthesia. 

### 2.3. Time to Onset of Paresthesia

We inferred dates of onset of paresthesia after 9/11/2001 by two methods: Among subjects with a baseline visit only, we used subjects’ self-reports of the number of months post-September 11, 2001 when they recalled first experiencing paresthesia of the lower extremities. For subjects who reported not having had paresthesia at their baseline enrollment interview but who then reported paresthesia at their first return visit, we estimated the date of onset of paresthesia as the midpoint between the dates of the baseline interview and the first return visit. We right-censored dates of onset of paresthesia of subjects in the following situations: (a) patients were considered as censored at initial baseline visit time for those who reported no paresthesia at the baseline visit and did not come back for a return visit; (b) patients who did not report paresthesia at both the baseline visit and return are considered as censored at return. 

### 2.4. Statistical Analyses

Descriptive statistics were used to summarize the characteristics of the study population. Categorical variables (or categorized continuous variables) were summarized using counts and percentages. Initial analyses compared persons with follow-up visits to those without follow-up visits. We used the chi-squared test to compare categorical variables of subjects with and without follow-up. We used population hazard functions for paresthesia-free survival and the log-rank test to evaluate differences in time to onset of paresthesia among subjects with versus those without a job related to cleaning WTC dust. For multivariable analyses, we used Cox proportional hazards regression analysis to calculate hazard ratios associated with various risk factors after adjusting for potential confounders. We performed Cox regression analyses both including and excluding those subjects who reported “occasional” paresthesia without ever reporting “often” or “almost continuous” paresthesia. All statistical analyses were performed using SAS version 9.4 (SAS Institute, Cary, NC, USA). 

## 3. Results

4165 subjects enrolled in the WTC EHC after January 1, 2008 when questions about paresthesia were first routinely included in the WTC EHC enrollment questionnaire. We excluded from our analyses, 11 subjects who were born after September 11, 2001, 132 subjects who reported paresthesia before September 11, 2001, and 458 subjects who reported diabetes, leaving 3564 individuals (Figure 1). After dropping subjects with diabetes, histories of cancer chemotherapy, paresthesia prior to 9/11 and those in utero on September 11, 2001, there remained for analysis data from 3411 individuals (Figure 1). 

The median date of enrollment was January 19, 2010 (interquartile range [IQR] = September 19, 2007–May 1, 2014). Of the 3564, 38.5% (1373/3564) returned for follow-up at a median of 35 months (IQR = 23–47 months) after enrollment. The proportions returning for return visits were associated with age, year of enrollment, and race/ethnicity (all *p* < 0.0001) (Table 1).

Among the 3411 subjects included in the present analyses, 50.7% were female, 48% were Caucasian, and 78% were non-Hispanic (Table 2). The median age was 53 (IQR = 44–61) years. At baseline, 605 (17.7%) reported feeling “often” or “almost continuous” symptoms of paresthesia of the lower extremities and these 605 were regarded as having had paresthesia at the number of months post-September 11, 2001 that each subject recalled having first noticed the symptom. The remaining 2806 (82.3%) were classified as not having paresthesia at baseline in our analysis. These 2806 subjects included 1004 (35.7%) who answered “occasionally” at either the baseline or return visit. 

### 3.1. Time to Onset of Paresthesia: Nonparametric Analysis 

To examine potential change in the hazard rate (or incidence rate) of paresthesia, we used nonparametric estimates of the Kaplan–Meier survival functions and the cumulative hazard functions for times-to-paresthesia after 9/11 in our study population. As is well known, the Kaplan–Meier survival functions are nonparametric estimates of the survival probabilities and the cumulative hazard function (Figure 2) is just the negative logarithm of the estimated survival functions. The nonparametric analysis is based on the 605 paresthesia cases reported at initial visit and the 159 new cases from M1. The above nonparametric cumulative hazard function of paresthesia-free survival suggests a long-term and continuing risk of paresthesia 17 years or more after the WTC disaster. More specifically, the slope for the cumulative hazard function in Figure 2 suggests a near-constant incidence rate of paresthesia in this population over the study period.

### 3.2. Time to Onset of Paresthesia: Multivariate Cox Regression 

Multivariable Cox proportional hazards regression analysis indicated a significantly elevated hazard ratio (HR = 1.369, *p* = 0.003) for paresthesia associated with having worked in a job that required cleaning of WTC dust after adjusting for age, race-ethnicity, depression, anxiety, post-traumatic stress disorder, and elevated body mass index (Table 3). These associations remained if we analyzed data only from the baseline questionnaire without considering data from the return visit. Blood glucose concentrations were not significant and thus did not enter into the model. 

Nonparametric hazard functions (Figure 3) comparing the 422 subjects who worked in a job that required cleaning of WTC dust (red curve) to the 2915 subjects who did not have such jobs (blue curve) showed a significant difference between the two groups (*p* < 0.0001, log-rank test). 

The nonparametric hazard curves in Figure 3 indicate significantly higher risk in time to onset of paresthesia among persons who worked in jobs that required cleaning of WTC dust in the workplace after adjusting for potential confounders. 

The Cox model shown in Table 3 categorized as paresthesia-free those subjects who reported paresthesia with frequencies of “none” or “occasional”. This might have weakened the observed statistical associations because some persons with “occasional” paresthesia might indeed have had the same condition as subjects with paresthesia of frequency “often” or “almost continuous”. We therefore analyzed the data excluding 1004 subjects with one or two reports of paresthesia of frequency “occasional” but who never reported paresthesia of frequency “often” or “almost continuous”. Cox regression analysis of the reduced data set (Table 4) showed significant associations of time to paresthesia with being employed in a job that required cleaning of WTC dust and with being heavily covered with dust on the day of the WTC disaster, September 11, 2001. The hazard ratio for working in a job that required cleaning of WTC dust increased from 1.35 (95% confidence interval CI = 1.11–1.69) in the model that included all subjects to 1.52 (95% CI = 1.24–1.87) in the model that excluded subjects with “occasional” paresthesia. 

## 4. Discussion

Among WTC survivors from the general community, we found significantly shorter time to onset of paresthesia among persons who worked in jobs that required cleaning of WTC dust in the workplace, or who were heavily covered with dust on the day of the disaster, after adjusting for potential confounders. Paresthesia was a common problem in this population, with 17.7% of subjects reporting symptoms of paresthesia at initial visit, even after excluding from the case definition those who reported paresthesias only of the upper extremities, and restricting the case definition to those who experienced the symptom with a frequency of “often” or “almost continuous”. The slope of the estimated cumulative hazard function was approximately constant, indicating a long latency period before onset of paresthesia among many subjects and a flat incidence rate in the WTC affected populations many years after 9/11.

Results of a previous analysis of risk factors for paresthesia among WTC EHC patients that did not take into account times to events were similar to those of one of the present models in finding significant associations of paresthesias with both working in jobs that required cleaning of WTC dust in the workplace and heavy exposure to WTC dust on September 11, 2001 after adjusting for potential confounders [13]. Of these two risk factors, only work in a job that required cleaning of WTC dust was significant in our initial Cox regression model that counted subjects with “occasional” paresthesia as paresthesia-free. The finding that being heavily covered with dust also was significant in the model that excluded subjects with “occasional” paresthesia suggests that exposures of some of the “occasional” paresthesia subjects were similar to those of subjects with “often” or “almost continuous” paresthesia. Categorization of subjects with “occasional” paresthesia as paresthesia-free during the study period appears to have reduced the statistical power of the analysis that included these subjects. 

The present findings cannot establish a causal relationship between WTC exposure and paresthesia. The findings are consistent, however, with the hypothesis that exposure to the WTC event during cleaning of dust in the workplace increased the subsequent risk of paresthesia and that the hazard continued to be relatively constant over extended times through approximately 17 years.

Our hazard function plots of survival times paresthesia-free suggest a long-term and continuing risk 17 years or more after the WTC disaster of paresthesia. Few instances of exposure to a time-limited event, including environmental disasters with neurotoxic consequences, have been reported and most exposures to neurotoxins result in symptoms within weeks or months [3,17,18]. In some instances, it is difficult to untangle whether a latency period precedes symptom expression of neuropathy or if a certain level of dose must be accumulated. Chemotherapy-associated neuropathy, for example, is more likely to develop in patients who have received larger doses over longer periods [19]. Occupational exposure to n-hexanes and low boiling point hydrocarbons have been associated with the development of a sub-acute neuropathy [20,21]. Removal of exposure of the occupational neurotoxins and cancer chemotherapy has been reported to improve neuropathic symptoms and in some cases also reverse the neuropathy or improve it [19,22]. The increased occurrence of symptoms over time might be due to other causes, including the onset of diabetes. Our available serum values did not suggest poor glycemic control in this population. While most patients fasted before blood donation, we did not record whose samples were provided after fasting and which were not. 

One conceptual model for our finding is that WTC exposure resulted in inhalation or ingestion of neurotoxins on airborne particulate matter and that these particulates became embedded in lung or other tissues, with slow leaching of the neurotoxins from these particles into the bloodstream over years, eventually leading to adequate bioaccumulation to cause expression of neurologic symptoms. Slow leaching of lead from retained bullet fragments has been associated with elevated blood lead levels [23]. Perhaps the delayed onset of symptoms in WTC subjects may be mediated by a delay in cumulative dose since the event, or may be due to a combination of industrial and environmental toxins related to the WTC disaster. 

Symptoms of paresthesia resolved or decreased in a subset of subjects between the enrollment and first return visits. This finding may be due to removal of the toxic exposure and subsequent reduction in symptoms. Another possibility may be due to medication for neuropathic symptoms with improvement in symptoms. Another consideration is that subjects started treatment for known etiologies of neuropathy that may have alleviated neuropathic symptoms. 

Hispanic ethnicity, older age, active mental health issues, and BMI also were associated with significantly elevated hazard ratios of paresthesia. Diabetes and pre-diabetes are the most common etiologies of neuropathy, specifically distal symmetric polyneuropathy of the lower extremities [24,25]. Although elevated BMI is associated with a greater risk of diabetes and pre-diabetes, serum glucose concentrations were not associated with time to paresthesia, suggesting that neither diabetes nor pre-diabetes were responsible for the reported paresthesias. 

Mood disorders are associated with different variants of neuropathy from diabetic, inflammatory, and chemotherapy-associated neuropathy [26,27,28]. The prevalence of depression is also greater in diabetics who have neuropathy [29,30]. The significant association of time to paresthesia with WTC exposure variables after controlling for covariates including measures of depression, anxiety, post-traumatic stress disorder and BMI, suggests a role of WTC-associated exposure. 

There are limitations to our study. We used a standard question for paresthesias that may not be specific for neuropathy and our question was not supplemented with complete neurological examinations, serum studies to search for etiologies of neuropathy, or additional investigations such as electromyography and nerve conduction study, or skin biopsy for evaluation of small fiber polyneuropathy. The questionnaire data also did not specifically address unilateral lower limb from bilateral limb paresthesia. Paresthesias due to other etiologies that would be considered independent of WTC exposure, such as unilateral lumbar radiculopathy, may have been included in our case definition. However, to improve the accuracy of our case definition, we only counted as cases those subjects who reported frequent symptoms. Counting subjects who reported paresthesia with frequency of “occasional” as paresthesia-free, however, likely weakened the statistical associations we observed compared with the model we created that excluded such subjects. We lacked a control group without exposure to the WTC disaster. Only 36% of subjects had follow-up visits. Whereas some subjects had two or more return visits during this period, the numbers were too small to warrant inclusion in this analysis and the subset of patients returning for earlier, follow-up visits might have been over-weighted with those with more severe symptoms. In addition, dates of onset of paresthesia were inferred from self-reports among subjects without return visits, and were calculated from the mid-points between visits for those with incident paresthesia at recall. Another limitation of the current study is that a portion of the reported paresthesia cases might be symptoms that are transient in nature. The patterns of symptom persistence and/or recurrence of paresthesia are worthy of further studies using more comprehensive follow up data. 

Paresthesia is a common symptom of neuropathy, but the estimated prevalence of neuropathy in the general population is only 2.4%–7% [31,32]. The higher prevalence of paresthesia in the present study sample strongly suggests that something about exposure to the WTC event or employment in jobs requiring cleaning of WTC dust increased the risk of neuropathy and paresthesia. 

## 5. Conclusions

Reports of paresthesia symptoms are common in the WTC EHC population. We also found significantly higher risk in time to onset of paresthesia for (a) working in a job that required cleaning of WTC dust in the workplace; and (b) being heavily exposed to WTC dust on September 11, 2001, after adjusting for potential confounders. This study suggests that WTC exposure is associated with the development of paresthesias and underscores the importance of further evaluating this population clinically to search for a possible mechanism.

## Figures and Tables

**Figure 1 ijerph-16-01429-f001:**
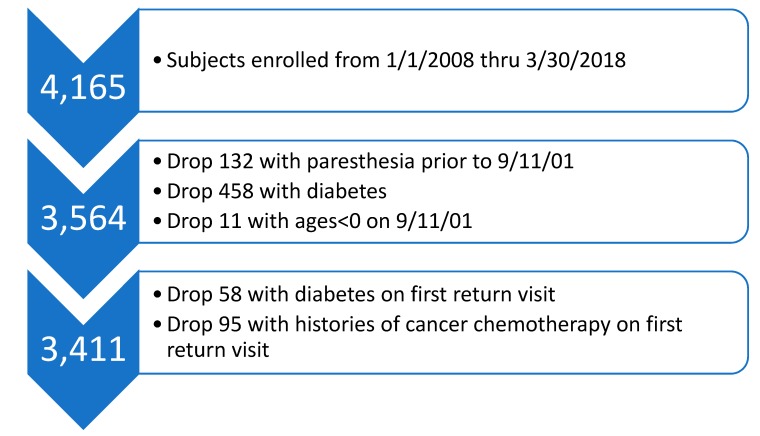
Flowchart of patients included in analyses of time to onset of paresthesia among patients of the Bellevue Hospital WTC (World Trade Center) EHC (Environmental Health Center).

**Figure 2 ijerph-16-01429-f002:**
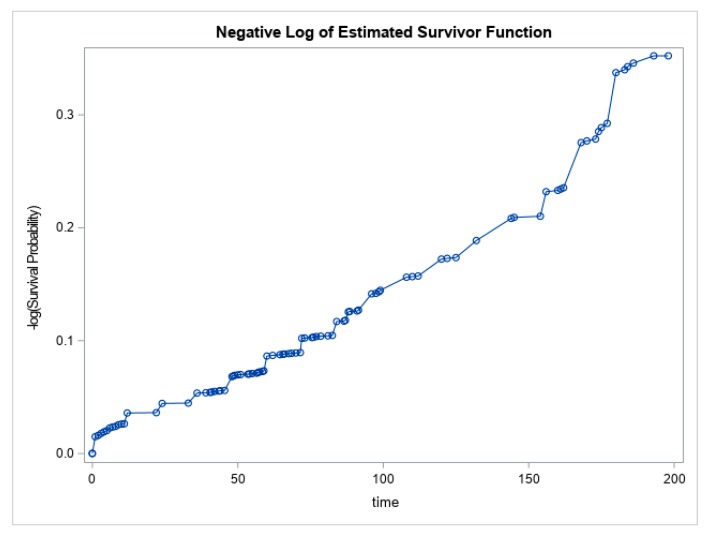
Nonparametric hazard function of paresthesia-free survival (*n* = 3411).

**Figure 3 ijerph-16-01429-f003:**
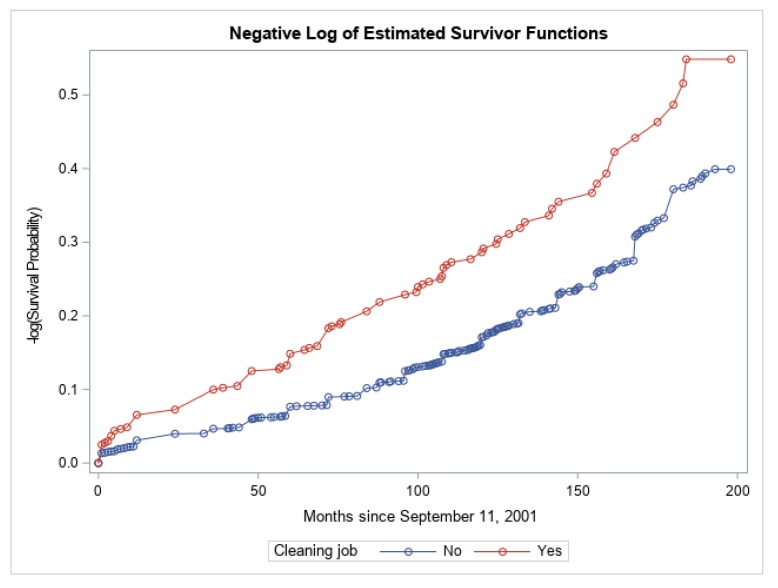
Nonparametric hazard functions of paresthesia-free survival stratified by cleaning job.

**Table 1 ijerph-16-01429-t001:** Baseline characteristics associated with returning percentage for a return visit (*n* = 3564).

Variable	Value	*N* at Baseline	*N* (%) with Return	*N* (%) without Return	*p*-Value
Gender					0.4
	Female	1811	711 (39.3)	1100 (60.7)	
	Male	1753	662 (37.8)	1091 (62.2)	
Age on 911					<0.0001
	<35	967	302 (31.2)	665 (68.8)	
	35–44	1155	452 (39.1)	703 (60.9)	
	45–54	1022	435 (42.6)	587 (57.4)	
	55–64	356	156 (43.8)	200 (56.2)	
	≥65	64	28 (43.8)	36 (56.2)	
Year of enrollment					<0.0001
	2008–2010	1432	746 (52.1)	686 (47.9)	
	2011–2013	817	381 (46.6)	436 (53.4)	
	2014–2015	538	197 (36.6)	341 (63.4)	
	2016–2018	777	49 (6.3)	728 (93.7)	
Race/Ethnicity					<0.0001
	Hispanic	799	382 (47.8)	417 (52.2)	
	Non-Hispanic Caucasian	1713	566 (33.0)	1147 (67.0)	
	Non-Hispanic African-American	705	294 (41.7)	411 (58.3)	
	Asian	263	93 (35.4)	170 (64.6)	
	Other or Native American	47	16 (34.0)	31 (66.0)	
	Missing race/ethnicity	37			

**Table 2 ijerph-16-01429-t002:** Characteristics of subjects at the baseline visit (*n* = 3411).

Variable	Value	*N* (Column %)
Gender		
	Male	1681 (49.3)
	Female	1730 (50.7)
Age at enrollment (years)		
	<25	138 (4.0)
	25–34	231 (6.8)
	35–44	552 (16.2)
	45–54	1006 (29.5)
	55–64	1001 (29.3)
	≥65	483 (14.2)
Median age at enrollment (IQR) 53(44–61)		
Year of enrollment	2008–2010	1372 (40.2)
	2011–2013	776 (22.8)
	2014–2015	499 (14.6)
	2016–2018	764 (22.4)
Race and ethnicity		
	Hispanic	765 (22.4)
	Non-Hispanic Caucasian	1635 (47.9)
	Non-Hispanic African- American	675 (19.8)
	Asian	258 (7.6)
	Other or Native American	44 (1.3)
	Missing	34 (1.0)
Subjects reported paresthesia	Yes	605 (17.7)
	No	2806 (82.3)

**Table 3 ijerph-16-01429-t003:** Multivariable Cox proportional hazards regression using both Baseline and Return Visit data (*n* = 3411).

Variable		Hazard Ratio	95% Confidence Interval		*p*-Value
Gender	F vs. M	1.14	0.99	1.33	0.08
Age on 911 (years)	Per unit increase	1.01	1.01	1.02	<0.0001
Race-Ethnicity (vs. Non-Hispanic Caucasian)	Hispanic	1.24	1.03	1.50	0.03
	Non-Hispanic African-American	1.12	0.92	1.37	0.2
	Asian	0.85	0.61	1.18	0.3
	Other or Native-American	1.83	1.07	3.13	0.03
Cleaning Job	Yes vs. No	1.37	1.11	1.69	0.003
Covered in Dust	Much vs. Little	1.09	0.94	1.27	0.2
Body mass index (kg/m^2^)	Per unit increase	1.02	1.01	1.03	0.001
Anxiety (vs. <1.75)	≥1.75	1.37	1.10	1.71	0.006
	Missing	1.45	0.28	7.58	0.7
Post-Traumatic Stress Disorder (vs. PCL*- < 44)	PCL ≥ 44	1.29	1.04	1.59	0.02
	Missing	0.98	0.19	5.14	0.9
Depression (vs. <1.75)	≥1.75	1.41	1.13	1.76	0.002
	Missing	1.45	0.28	7.58	0.7

* PCL is Post-Traumatic Stress Disorder Check List.

**Table 4 ijerph-16-01429-t004:** Multivariable Cox proportional hazards regression using Baseline and Return Visit data, excluding subjects with “occasional” paresthesia (*n* = 2407).

Variable		Hazard Ratio	95% Confidence Interval		*p*-Value
Gender	F vs. M	1.22	1.05	1.41	0.01
Age on 911 (years)	Per unit increase	1.01	1.01	1.02	<0.0001
Race-Ethnicity (vs. Non-Hispanic Caucasian)	Hispanic	0.85	0.61	1.18	0.3
	Non-Hispanic African-American	1.32	1.09	1.59	0.004
	Asian	1.29	1.06	1.57	0.0106
	Other or Native-American	1.75	1.02	2.99	0.04
Cleaning Job	Yes vs. No	1.52	1.24	1.87	<0.0001
Covered in Dust	Much vs. Little	1.19	1.03	1.38	0.02
Body mass index (kg/m^2^)	Per unit increase	1.02	1.01	1.03	0.0007
Anxiety (vs. <1.75)	≥1.75	1.51	1.21	1.89	0.0002
	Missing	1.10	0.25	4.82	0.9
Post-Traumatic Stress Disorder (vs. PCL < 44)	PCL ≥ 44	1.35	1.09	1.67	0.006
	Missing	1.42	0.32	6.27	0.6
Depression (vs. <1.75)	≥1.75	1.49	1.19	1.85	0.0004
	Missing	1.10	0.25	4.82	0.9

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
