# Peer review of "Time to Onset of Paresthesia Among Community Members Exposed to the World Trade Center Disaster"

_ijerph, 2019, doi:10.3390/ijerph16081429_

Round 1
Reviewer 1 Report
This is a very interesting manuscript. Valuable for publication.
Some remarks however, so this manuscript can be even better than the present version.
Some remarks can help in further research.
Statistical analysis: to be reviewed by an expert.
The authors are encouraged to write a review on all the medical findings in WTC-workers/survivors.
Introduction:
The authors did refer to an article of Stecker et al (ref 14). The same author also published an interesting article on the effect of WTC dust on rat sciatic nerve. This is worthwhile to be referred to in this manuscript.
J Occup Environ Med. 2014 Oct;56(10):1024-8. doi: 10.1097/JOM.0000000000000296.
Analysis of short-term effects of World Trade Center dust on rat sciatic nerve.
Stecker M1, Segelnick J, Wilkenfeld M.OBJECTIVE:
The purpose of this study was to investigate the short-term effects of residual dust from the World Trade Center (WTC) on rat sciatic nerve.
METHODS:
Nerve action potentials were recorded in nerves exposed to dust from the WTC as well as control nerves.
RESULTS:
There was a reduction in the conduction velocity of nerves exposed to a high concentration of the dust from the WTC when compared with controls.
CONCLUSIONS:
Although there are statistically significant reductions in conduction velocity when exposed to the WTC dust in this pilot study, additional studies both clinical and basic will be needed to further understand the significance of these results.
Our review on neurological disease after terrorist attacks also is commenting on neuropathies after WTC, sarin intoxication (sarin attacks in Tokyo): maybe also valuable to be referred to.
Acta Neurol Belg. 2018 Jun;118(2):193-199. doi: 10.1007/s13760-018-0924-x. Epub 2018 Apr 24.
Neurological disease in the aftermath of terrorism: a review.
De Cauwer H1,2, Somville FJMP3,4,5.
The purpose of our review is to discuss current knowledge on long-term sequelae and neurological disorders in the aftermath of a terrorist attack. The specific aspects of both psychological and physical effects are mentioned in more detail in this review. Also, the outcomes such as stress-related disorders, cardiovascular disease, and neurodegenerative disease are explained. Moreover, PTSD and posttraumatic structural brain changes are a topic for further investigations of the patients suffering from these attacks. Not only the direct victims are prone to the after effects of the terroristic attacks, but the rescue workers, physicians, witnesses and worldwide citizens may also be affected by PTSD and other neurological diseases as well. The determination of a whole series of risk factors for developing neurological disorders can be a means to set up early detection, preventative measures, to refine treatment and thus to gain better outcome in the future.
3. other authors also stress the toxin theory in neurological disease in the military: exposure to toxic agents due to burning oil fields (similar to full kerosine tanks of the airplanes entering the WTC), low radioactive uranium during Desert Storm and neurological diseases or after sarin attacks in Japan.
Miyaki K, Nishiwaki Y, Maekawa K, Ogawa Y, Asukai N, Yoshimura K, Etoh N, Matsumoto Y, Kikuchi Y, Kumagai N, Omae K. Effects of sarin on the nervous system of subway workers seven years after the Tokyo subway sarin attack. J Occup Health. 2005;47(4):299-304.
Jones E. Historical approaches to post-combat disorders. Philos Trans R Soc Lond B Biol Sci. 2006;361(1468):533-542.
Beard JD, Kamel F. Military service, deployments, and exposures in relation to amyotrophic lateral sclerosis etiology and survival. Epidemiol Rev. 2015;37:55-70.
Barth SK, Kang HK, Bullman TA, Wallin MT. Neurological mortality among U.S. veterans of the Persian Gulf War: 13-year follow-up. Am J Ind Med. 2009;52(9):663-670.
Tai H, Cui L, Shen D, Li D, Cui B, Fang J. Military service and the risk of amyotrophic lateral sclerosis: A meta-analysis. J Clin Neurosci. 2017;45:337-342.
Engdahl BE, James LM, Miller RD, Leuthold AC, Lewis SM, Carpenter AF, Georgopoulos AP. A Magnetoencephalographic (MEG) Study of Gulf War Illness (GWI). EBioMedicine. 2016;12:127-132.
Methods:
The DN4 neuropathic pain questionnary is more liable and perhaps should be used in further study in a smaller sample with also EMG/nerve conduction study (if you need a neurologist to perform the nerve conduction tests, you can give me a call).
This article about DN4 also shows an association between neuropathic pain and depression: maybe to be referred to in the present manuscript.
Clin J Pain. 2018 Jan;34(1):30-36. doi: 10.1097/AJP.0000000000000512.
Sensitivity of the DN4 in Screening for Neuropathic Pain Syndromes.
VanDenKerkhof EG1, Stitt L2, Clark AJ3, Gordon A4, Lynch M5, Morley-Forster PK6, Nathan HJ7, Smyth C7, Toth C8, Ware MA9, Moulin DE10.
OBJECTIVES:
Several tools have been developed to screen for neuropathic pain. This study examined the sensitivity of the Douleur Neuropathique en 4 Questions (DN4) in screening for various neuropathic pain syndromes.
MATERIALS AND METHODS:
This prospective observational study was conducted in 7 Canadian academic pain centers between April 2008 and December 2011. All newly admitted patients (n=2199) were approached and 789 eligible participants form the sample for this analysis. Baseline data included demographics, disability, health-related quality of life, and pain characteristics. Diagnosis of probable or definite neuropathic pain was on the basis of history, neurological examination, and ancillary diagnostic tests.
RESULTS:
The mean age of study participants was 53.5 years and 54.7% were female; 83% (n=652/789) screened positive on the DN4 (≥4/10). The sensitivity was highest for central neuropathic pain (92.5%, n=74/80) and generalized polyneuropathies (92.1%, n=139/151), and lowest for trigeminal neuralgia (69.2%, n=36/52). After controlling for confounders, the sensitivity of the DN4 remained significantly higher for individuals with generalized polyneuropathies (odds ratio [OR]=4.35; 95% confidence interval [CI]: 2.15, 8.81), central neuropathic pain (OR=3.76; 95% CI: 1.56, 9.07), and multifocal polyneuropathies (OR=1.72; 95% CI: 1.03, 2.85) compared with focal neuropathies.
DISCUSSION:
The DN4 performed well; however, sensitivity varied by syndrome and the lowest sensitivity was found for trigeminal neuralgia. A positive DN4 was associated with greater pain catastrophizing, disability and anxiety/depression, which may be because of disease severity, and/or these scales may reflect magnification of sensory symptoms and findings. Future research should examine how the DN4 could be refined to improve its sensitivity for specific neuropathic pain conditions.
Peripheral ankle edema is also a factor that can induce paresthesias due to pressure neuropathy. Maybe also to be included in further study.
Discussion
Table 3 and discussion: the exposure of cleaning workers to WTC is very much higher since cleaning up ‘ground zero’ took a year or more. This in contrast to people working in the WTC / vicinity, that have a much shorter exposure time. So cleaning job is off course a larger hazard.
Slow leaching from neurotoxins from particles embedded in the lungs: this migth be right, and is similar to other reports on lead intoxication from bullet fragments.
MMWR Morb Mortal Wkly Rep. 2017 Feb 10;66(5):130-133. doi: 10.15585/mmwr.mm6605a2.
Elevated Blood Lead Levels Associated with Retained Bullet Fragments - United States, 2003-2012.
Weiss D, Tomasallo CD, Meiman JG, Alarcon W, Graber NM, Bisgard KM, Anderson HA.
BMI and respiratory problems and depression are also linked to OSAS which had also been linked to PTSD. On the other hand, OSAS has also been linked to depression and solvent exposure.
Ayappa I, Sunderram J, Black K, Twumasi A, Udasin I, Harrison D, Carson JL, Lu SE, Rapoport DM. A comparison of CPAP and CPAPFLEX in the treatment of obstructive sleep apnea in World Trade Center responders: study protocol for a randomized controlled trial.
Trials. 2015 Sep 10;16:403.
Viaene M1, Vermeir G, Godderis L. Sleep disturbances and occupational exposure to solvents. Sleep Med Rev. 2009 Jun;13(3):235-43. doi: 10.1016/j.smrv.2008.07.003. Epub 2009 Feb 7.
This recent article also stresses the effect of 9/11 on the recovery workers. The present manuscript of Thawani et al demonstrates they are not forgotten.
Prehosp Disaster Med. 2018 Aug;33(4):436-440. doi: 10.1017/S1049023X1800064X.
The Forgotten Responders: The Ongoing Impact of 9/11 on the Ground Zero Recovery Workers. Smith EC1, Burkle FM2.
Author Response
This is a very interesting manuscript. Valuable for publication.
Some remarks however, so this manuscript can be even better than the present version.
Some remarks can help in further research.
Statistical analysis: to be reviewed by an expert.
The authors are encouraged to write a review on all the medical findings in WTC-workers/survivors.
The authors did refer to an article of Stecker et al (ref 14). The same author also published an interesting article on the effect of WTC dust on rat sciatic nerve. This is worthwhile to be referred to in this manuscript. J Occup Environ Med. 2014 Oct;56(10):1024-8. doi: 10.1097/JOM.0000000000000296. Analysis of short-term effects of World Trade Center dust on rat sciatic nerve. Stecker M1 , Segelnick J, Wilkenfeld M.
OBJECTIVE: The purpose of this study was to investigate the short-term effects of residual dust from the World Trade Center (WTC) on rat sciatic nerve. METHODS: Nerve action potentials were recorded in nerves exposed to dust from the WTC as well as control nerves. RESULTS: There was a reduction in the conduction velocity of nerves exposed to a high concentration of the dust from the WTC when compared with controls. CONCLUSIONS: Although there are statistically significant reductions in conduction velocity when exposed to the WTC dust in this pilot study, additional studies both clinical and basic will be needed to further understand the significance of these results.
Response: Thank you we have included the Stecker reference in the Introduction in lines 40-42.
Our review on neurological disease after terrorist attacks also is commenting on neuropathies after WTC, sarin intoxication (sarin attacks in Tokyo): maybe also valuable to be referred to. Acta Neurol Belg. 2018 Jun;118(2):193-199. doi: 10.1007/s13760- 018-0924-x. Epub 2018 Apr 24. Neurological disease in the aftermath of terrorism: a review. De Cauwer H 1,2 , Somville FJMP 3,4,5 . The purpose of our review is to discuss current knowledge on long-term sequelae and neurological disorders in the aftermath of a terrorist attack. The specific aspects of both psychological and physical effects are mentioned in more detail in this review. Also, the outcomes such as stress-related disorders, cardiovascular disease, and neurodegenerative disease are explained
Moreover, PTSD and posttraumatic structural brain changes are a topic for further investigations of the patients suffering from these attacks. Not only the direct victims are prone to the after effects of the terroristic attacks, but the rescue workers, physicians, witnesses and worldwide citizens may also be affected by PTSD and other neurological diseases as well. The determination of a whole series of risk factors for developing neurological disorders can be a means to set up early detection, preventative measures, to refine treatment and thus to gain better outcome in the future. 3.
3. other authors also stress the toxin theory in neurological disease in the military: exposure to toxic agents due to burning oil fields (similar to full kerosine tanks of the airplanes entering the WTC), low radioactive uranium during Desert Storm and neurological diseases or after sarin attacks in Japan. Miyaki K, Nishiwaki Y, Maekawa K, Ogawa Y, Asukai N, Yoshimura K, Etoh N, Matsumoto Y, Kikuchi Y, Kumagai N, Omae K. Effects of sarin on the nervous system of subway workers seven years after the Tokyo subway sarin attack. J Occup Health. 2005;47(4):299-304. Jones E. Historical approaches to post-combat disorders. Philos Trans R Soc Lond B Biol Sci. 2006;361(1468):533-542. 4/1/2019 MDPI | Reply review report https://susy.mdpi.com/user/manuscripts/review/7789053?report=4013071 3/5 Beard JD, Kamel F. Military service, deployments, and exposures in relation to amyotrophic lateral sclerosis etiology and survival. Epidemiol Rev. 2015;37:55-70. Barth SK, Kang HK, Bullman TA, Wallin MT. Neurological mortality among U.S. veterans of the Persian Gulf War: 13-year follow-up. Am J Ind Med. 2009;52(9):663-670. Tai H, Cui L, Shen D, Li D, Cui B, Fang J. Military service and the risk of amyotrophic lateral sclerosis: A meta-analysis. J Clin Neurosci. 2017;45:337- 342. Engdahl BE, James LM, Miller RD, Leuthold AC, Lewis SM, Carpenter AF, Georgopoulos AP. A Magnetoencephalographic (MEG) Study of Gulf War Illness (GWI). EBioMedicine. 2016;12:127-132.
Response: We have added the citation to sarin exposure since it is associated with peripheral neuropathic symptoms. Please see Page 1 line 38 continuing into Page 2 line 4 of the Introduction section. We have also referenced Acta Neurol Belg. 2018 Jun;118(2):193-199 and Am J Ind Med. 2009;52(9):663-670 in the Discussion section see Page 8, line 19.
We have not included references in the Introduction regarding PTSD and stress since these references are related to cerebrovascular events, headaches and motor neuron disease which have different pathologic mechanisms than paresthesia.
Methods: The DN4 neuropathic pain questionnaire is more liable and perhaps should be used in further study in a smaller sample with also EMG/nerve conduction study (if you need a neurologist to perform the nerve conduction tests, you can give me a call). This article about DN4 also shows an association between neuropathic pain and depression: maybe to be referred to in the present manuscript. Clin J Pain. 2018 Jan;34(1):30-36. doi: 10.1097/AJP.0000000000000512. Sensitivity of the DN4 in Screening for Neuropathic Pain Syndromes. VanDenKerkhof EG1 , Stitt L 2 , Clark AJ 3 , Gordon A 4 , Lynch M5 , MorleyForster PK 6 , Nathan HJ 7 , Smyth C 7 , Toth C 8 , Ware MA 9 , Moulin DE 10 . OBJECTIVES: Several tools have been developed to screen for neuropathic pain. This study examined the sensitivity of the Douleur Neuropathique en 4 Questions (DN4) in screening for various neuropathic pain syndromes.
MATERIALS AND METHODS: This prospective observational study was conducted in 7 Canadian academic pain centers between April 2008 and December 2011. All newly admitted patients (n=2199) were approached and 789 eligible participants form the sample for this analysis. Baseline data included demographics, disability, health-related quality of life, and pain characteristics. Diagnosis of probable or definite neuropathic pain was on the basis of history, neurological examination, and ancillary diagnostic tests. RESULTS: The mean age of study participants was 53.5 years and 54.7% were female; 83% (n=652/789) screened positive on the DN4 (≥4/10). The sensitivity was highest for central neuropathic pain (92.5%, n=74/80) and generalized polyneuropathies (92.1%, n=139/151), and lowest for trigeminal neuralgia (69.2%, n=36/52). After controlling for confounders, the sensitivity of the DN4 remained significantly higher for individuals with generalized polyneuropathies (odds ratio [OR]=4.35; 95% confidence interval [CI]: 2.15, 8.81), central neuropathic pain (OR=3.76; 95% 4/1/2019 MDPI | Reply review report https://susy.mdpi.com/user/manuscripts/review/7789053?report=4013071 4/5 CI: 1.56, 9.07), and multifocal polyneuropathies (OR=1.72; 95% CI: 1.03, 2.85) compared with focal neuropathies. DISCUSSION: The DN4 performed well; however, sensitivity varied by syndrome and the lowest sensitivity was found for trigeminal neuralgia. A positive DN4 was associated with greater pain catastrophizing, disability and anxiety/depression, which may be because of disease severity, and/or these scales may reflect magnification of sensory symptoms and findings. Future research should examine how the DN4 could be refined to improve its sensitivity for specific neuropathic pain conditions. Peripheral ankle edema is also a factor that can induce paresthesias due to pressure neuropathy. Maybe also to be included in further study.
Response: Thank you for these interesting remarks. These could be useful in the conduct of future studies of patients with neuropathic symptoms.
Discussion Table 3 and discussion: the exposure of cleaning workers to WTC is very much higher since cleaning up ‘ground zero’ took a year or more. This in contrast to people working in the WTC / vicinity, that have a much shorter exposure time. So cleaning job is off course a larger hazard. Slow leaching from neurotoxins from particles embedded in the lungs: this migth be right, and is similar to other reports on lead intoxication from bullet fragments. MMWR Morb Mortal Wkly Rep. 2017 Feb 10;66(5):130-133. doi: 10.15585/mmwr.mm6605a2. Elevated Blood Lead Levels Associated with Retained Bullet Fragments - United States, 2003-2012. Weiss D, Tomasallo CD, Meiman JG, Alarcon W, Graber NM, Bisgard KM, Anderson HA. BMI and respiratory problems and depression are also linked to OSAS which had also been linked to PTSD. On the other hand, OSAS has also been linked to depression and solvent exposure. Ayappa I, Sunderram J, Black K, Twumasi A, Udasin I, Harrison D, Carson JL, Lu SE, Rapoport DM. A comparison of CPAP and CPAPFLEX in the treatment of obstructive sleep apnea in World Trade Center responders: study protocol for a randomized controlled trial. Trials. 2015 Sep 10;16:403. Viaene M1 , Vermeir G, Godderis L. Sleep disturbances and occupational exposure to solvents. Sleep Med Rev. 2009 Jun;13(3):235
Response: Thank you for the reference on bullet fragments, which we included in the Discussion (page 8, line 35). We do not believe the PTSD or sleep apnea studies to be relevant to our manuscript and have not included them..

Reviewer 2 Report
The manuscript ijerph-472307 entitled ‘Times-to-Onset of Paresthesia among Community Members Exposed to the World Trade Center Disaster’ by Thawani et al. examined the time to onset for paresthesias was associated with, and an indicator of toxicant dust exposure at the World Trade Center. The authors make a compelling case for correlation of exposure to the altered dermal responses associated with paresthesia. The project, data, analysis as well as the conclusions are well laid out and described. The modeling system is well explained and I see no obvious concerns with the methodology or the analysis. There are only some very minor structural issues that can be addressed prior to further consideration for publication.
Page 1
Line 2 – in the title, is the term “Times-to-Onset’ appropriate or should it be “Time-to-Onset”?
Line 23 – change ‘appears’ to ‘is’
Line 30 – should ‘local’ be included, it seems that would be implied, but if not, the double local is redundant can be rewritten as ‘local workers and residents, as well as other affected community members’
Page 2
Line 4 – ‘the severity’
Line 8-10 – the sentence ‘In addition, paresthesias were associated with later (2013-2015) compared to earlier (2008-2010) year of enrollment in the WTC EHC suggesting a long latency period between exposure and development of symptoms in some patients’ seems a bit awkward. Consider rewriting to ‘Paresthesias were associated with later (2013-2015) enrollment in the WTC EHC suggesting a long latency period between exposure and development of symptoms in some patients’ – the comparison should be understood regarding the year of enrollment.
Line 13-14 – reverse ‘to further describe’ to ‘to describe further’
Line 33 – ‘the use’
Line 35 – delete ‘In the present statistical analyses,’ and start with ‘We investigated…’ It is implied that authors are discussing the current analysis.
Line 47 – change ‘the date of the baseline interview and the date of the first return visit’ to ‘the dates of the baseline interview and the first return visit’
Page 3
Line 1 – insert ‘the’ before ‘initial’ and before the second ‘baseline’
Line 5 – insert ‘the’ before ‘characteristics’
Line 21 – change ‘there remained for analysis data from 3,411 individuals’ to ‘there remained 3,411 individuals included for further analysis’
Line 22-23 – FIGURE 1 – ‘chemotherapy’ is misspelled in the Drop 95 line
Page 4
Line 12 – insert ‘the’ before baseline in Table 2 header
Page 5
Line 10-11 – ‘approximately a constant’ is confusing – it’s either a constant or not. This should be re-worded
Line 15 – change ‘Times’ to ‘Time’
Page 6
Table 3 – there is a “Non-Hispanic” that needs a hyphen
Figure 3 legend – remove the hyphen from ‘cleaning job’
Line 13 – change ‘shorter times’ to ‘shorter time’
Page 7
Line 5 – consider changing ‘heavy’ to ‘significant’ or ‘substantial’
Line 19 – the ending of the sentence ‘periods of time’ could just be ‘periods’
Line 30 – the sentence ‘eventually leading to a high enough cumulative doses to nerves to elicit neurologic symptoms.’ Is a little awkward. Partly plural and use of nerves and neurological is redundant. Consider ‘eventually leading to bioaccumulation which elicits neurologic symptoms.’
Line 46-47 – reverse ‘also is’ to ‘is also’
Page 8
Line 7 – remove ‘This’ and replace with ‘Clustering ‘occasional’ with those without paresthesia may have weakened the statistical association observed’. If that was the case – why do that? Was the data analyzed with the groups unclustered?
Line 9 – ‘dose’ may not be the best word to use – maybe ‘concentration’?
Line 12 – delete ‘second’ – the phrase ‘follow-up’ implies second visit
Author Response
The manuscript ijerph-472307 entitled ‘Times-to-Onset of Paresthesia among Community Members Exposed to the World Trade Center Disaster’ by Thawani et al. examined the time to onset for paresthesias was associated with, and an indicator of toxicant dust exposure at the World Trade Center. The authors make a compelling case for correlation of exposure to the altered dermal responses associated with paresthesia. The project, data, analysis as well as the conclusions are well laid out and described. The modeling system is well explained and I see no obvious concerns with the methodology or the analysis. There are only some very minor structural issues that can be addressed prior to further consideration for publication.
Page 1
Line 2 – in the title, is the term “Times-to-Onset’ appropriate or should it be “Time-to-Onset”?
Response: We thank the reviewer for the suggestion. We agree that “Time-to-onset” is better, but have decided after a fast review of the literature that the hyphens are not appropriate unless the term is used as a composite adjective. We have thus changed “times-to-onset” to “times to onset,” and similarly changed “times-to-paresthesia” to “times to paresthesia.”
Line 23 – change ‘appears’ to ‘is’
Line 30 – should ‘local’ be included, it seems that would be implied, but if not, the double local is redundant can be rewritten as ‘local workers and residents, as well as other affected community members’
Page 2
Line 4 – ‘the severity’
Line 8-10 – the sentence ‘In addition, paresthesias were associated with later (2013-2015) compared to earlier (2008-2010) year of enrollment in the WTC EHC suggesting a long latency period between exposure and development of symptoms in some patients’ seems a bit awkward. Consider rewriting to ‘Paresthesias were associated with later (2013-2015) enrollment in the WTC EHC suggesting a long latency period between exposure and development of symptoms in some patients’ – the comparison should be understood regarding the year of enrollment.
Line 13-14 – reverse ‘to further describe’ to ‘to describe further’
Line 33 – ‘the use’
Line 35 – delete ‘In the present statistical analyses,’ and start with ‘We investigated…’ It is implied that authors are discussing the current analysis.
Line 47 – change ‘the date of the baseline interview and the date of the first return visit’ to ‘the dates of the baseline interview and the first return visit’
Page 3
Line 1 – insert ‘the’ before ‘initial’ and before the second ‘baseline’
Line 5 – insert ‘the’ before ‘characteristics’
Response: We accept each of the above revisions.
Line 21 – change ‘there remained for analysis data from 3,411 individuals’ to ‘there remained 3,411 individuals included for further analysis’
Response: We reject this change as “included” seems not to have utility in the suggested revision.
Line 22-23 – FIGURE 1 – ‘chemotherapy’ is misspelled in the Drop 95 line
Page 4
Line 12 – insert ‘the’ before baseline in Table 2 header
Page 5
Line 10-11 – ‘approximately a constant’ is confusing – it’s either a constant or not. This should be re-worded
Line 15 – change ‘Times’ to ‘Time’
Page 6
Table 3 – there is a “Non-Hispanic” that needs a hyphen
Response: We accept each of the above suggestions and have made the recommended revisions.
Figure 3 legend – remove the hyphen from ‘cleaning job’
Response: We have made this revision, but have not been able to highlight the change in the legend.
Line 13 – change ‘shorter times’ to ‘shorter time’
Response: As with the suggested change to “times-to-onset,” we reject this suggestion.
Page 7
Line 5 – consider changing ‘heavy’ to ‘significant’ or ‘substantial’
Response: We appreciate the suggestion, but prefer the current wording.
Line 19 – the ending of the sentence ‘periods of time’ could just be ‘periods’
Response: We accept this suggestion.
Line 30 – the sentence ‘eventually leading to a high enough cumulative doses to nerves to elicit neurologic symptoms.’ Is a little awkward. Partly plural and use of nerves and neurological is redundant. Consider ‘eventually leading to bioaccumulation which elicits neurologic symptoms.’
Response: We have modified the suggestion to, “to adequate bioaccumulation to cause expression of neurologic symptoms.”
Line 46-47 – reverse ‘also is’ to ‘is also’
Page 8
Line 7 – remove ‘This’ and replace with ‘Clustering ‘occasional’ with those without paresthesia may have weakened the statistical association observed’. If that was the case – why do that? Was the data analyzed with the groups unclustered?
Response: Thank you for this suggestion. In response, we have added an analysis excluding subjects whose frequency of paresthesia was at most “occasional.” The findings are of interest finding because Cox modeling of this reduced data set showed significant hazard ratios for both being employed in a cleaning job and being heavily covered with dust on 9/11/2001. We have thus modified the sentence in the Discussion (along with the previous sentence) and have added a new table, Table 4, showing these results. We also have modified the Abstract, the first paragraph of the Discussion, and the Conclusion. We also deleted the second sentence of the second paragraph of the Discussion, as this sentence had discussed possible reasons for the difference between the present findings and those of our earlier logistic regression analysis. The two sets of findings – those from logistic regression and those from Cox regression -- now agree, at least after excluding the “occasional” paresthesia subjects from those counted as never having had paresthesia.
Line 9 – ‘dose’ may not be the best word to use – maybe ‘concentration’?
Response: The point of this sentence is unclear, so we have deleted the sentence.
Line 12 – delete ‘second’ – the phrase ‘follow-up’ implies second visit
Response: We agree and have made the change.